# Simple Clinical Screening Underestimates Malnutrition in Surgical Patients with Inflammatory Bowel Disease—An ACS NSQIP Analysis

**DOI:** 10.3390/nu14050932

**Published:** 2022-02-22

**Authors:** Mohamed A. Abd-El-Aziz, Martin Hübner, Nicolas Demartines, David W. Larson, Fabian Grass

**Affiliations:** 1Department of Internal Medicine, Texas Tech University Health Sciences Center, Paul L. Foster School of Medicine, El Paso, TX 79905, USA; mohamed.abd-el-maksoud@ttuhsc.edu; 2Division of Colon and Rectal Surgery, Department of Surgery, Mayo Clinic, 200 First Street SW, Rochester, MN 55905, USA; 3Department of Visceral Surgery, Lausanne University Hospital CHUV, University of Lausanne (UNIL), Bugnon 46, 1011 Lausanne, Switzerland; martin.hubner@chuv.ch (M.H.); demartines@chuv.ch (N.D.); fabian.grass@chuv.ch (F.G.)

**Keywords:** malnutrition, inflammatory bowel disease, Crohn’s disease, ulcerative colitis, albumin, surgery

## Abstract

The present large scale study aimed to assess the prevalence and consequences of malnutrition, based on clinical assessment (body mass index and preoperative weight loss) and severe hypoalbuminemia (<3.1 g/L), in a representative US cohort undergoing IBD surgery. The American College of Surgeons National Quality improvement program (ACS-NSQIP) Public User Files (PUF) between 2005 and 2018 were assessed. A total of 25,431 patients were identified. Of those, 6560 (25.8%) patients had severe hypoalbuminemia, 380 (1.5%) patients met ESPEN 2 criteria (≥10% weight loss over 6 months PLUS BMI < 20 kg/m^2^ in patients <70 years OR BMI < 22 kg/m^2^ in patients ≥70 years), and 671 (2.6%) patients met both criteria (severe hypoalbuminemia and ESPEN 2). Patients who presented with malnutrition according to any of the three definitions had higher rates of overall, minor, major, surgical, and medical complications, longer LOS, higher mortality and higher rates of readmission and reoperation. The simple clinical assessment of malnutrition based on BMI and weight loss only, considerably underestimates its true prevalence of up to 50% in surgical IBD patients and calls for dedicated nutritional assessment.

## 1. Introduction

Infectious complications or medically refractory disease are surgical indications in patients with inflammatory bowel disease (IBD), emphasizing the importance of preoperative optimization strategies, if feasible, to improve surgical outcomes [1,2]. Hence, proper nutritional assessment is mandatory to identify patients at risk and to launch preoperative nutritional support [3,4]. Hypoalbuminemia, as a marker of preoperative systemic inflammation, is associated with intra-abdominal septic complications and may help to identify patients at increased risk [5,6]. Body mass index (BMI) and weight loss are among the screening tools suggested by the European Society for Clinical Nutrition and Metabolism (ESPEN) [7]. However, clinical assessment solely based on these tools may underestimate the true prevalence of malnutrition and hence, entail nutritional undertreatment in IBD patients. This may especially apply to the US population facing an increasing prevalence of obesity in both the general population and IBD patients, with up to half of patients being either overweight or obese [8].

The present large scale study aimed to assess the prevalence and consequences of malnutrition, based on both clinical assessment and severe hypoalbuminemia, in a representative US cohort undergoing IBD surgery.

## 2. Materials and Methods

### 2.1. Data

The American College of Surgeons National Quality improvement program (ACS-NSQIP) Public User Files (PUF) between 2005 and 2018 were assessed. The ACS-NSQIP is an externally validated and outcome-based database that was initially created for quality improvement purposes. Each participant center has trained data abstractors who collect surgical data based on standardized definitions. Those data include demographics, anthropometrics, perioperative and post-operative details. The final pooled PUF represent 20% of surgical patients in the US.

### 2.2. Cohort

All adult patients who underwent surgery for ulcerative colitis (UC) or Crohn’s disease (CD) between 2005 and 2018, and were reported in the ACS-NSQIP PUF, were included in the analysis. Current procedure terminology and international classification of disease codes were used to select patients as specified in Figure 1.

The patients with missing anthropometric data, albumin level, or who had albumin measured more than 21 days before the index surgery, ASA class 5, age < 18 years old, preoperative ascites, or a history of esophageal varices were excluded from the analysis. Further excluded were patients who were on hemodialysis at the time of surgery, patients who were ventilator-dependent or in coma >24 h preoperatively, or patients who had pneumonia at the time of surgery.

### 2.3. Assessment of Malnutrition

The patients were divided into four mutually exclusive, nonoverlapping groups: (1) patients with preoperative albumin levels < 3.1 g/L which we referred to as severe hypoalbuminemia, (2) patients who had clinical parameters for malnutrition according to the ESPEN definition (≥10% weight loss over 6 months PLUS BMI < 20 kg/m^2^ in patients <70 years OR BMI < 22 kg/m^2^ in patients ≥70 years) which we referred to as ESPEN 2 criteria [9], (3) patients who fulfilled both of the aforementioned criteria, and (4) patients who had neither of the aforementioned criteria. Of note, body composition and muscle mass are not reported in the ACS-NSQIP and, therefore, the ESPEN 2 group does not meet the full definition of clinical malnutrition as suggested by the ESPEN guidelines.

### 2.4. Covariates and Outcomes

Baseline demographics, preoperative laboratories (hematocrit, platelets, liver function tests, including serum glutamic oxaloacetic transaminase and international normalized ratio) and surgical details (approach, surgical setting (elective vs. emergency), extent of resection, diversion at time of surgery) were compared between the four cohorts. The primary outcomes were the prevalence of malnutrition, according to the aforementioned definitions, overall and within both disease categories (UC and CD). The secondary outcomes included 30 day complications as defined by the standardized ACS-NSQIP definitions [10].

Surgical complications included any surgical site infection (SSI, superficial, deep or organ space), wound disruption, systemic sepsis (sepsis or septic shock), or the need for blood transfusion. Medical complications were defined as renal complications (progressive renal failure and/or acute kidney injury), respiratory complications (pneumonia, unplanned intubation, and/or on a mechanical ventilator ≥48 h), major adverse cardiovascular events (MACE: stroke, cardiac arrest requiring cardiopulmonary resuscitation, and/or myocardial infarction), and vascular thromboembolism (VTE: pulmonary embolism and/or deep venous thrombosis). Minor complications included UTI and superficial SSI. Major complications included myocardial infarction, cardiac arrest requiring cardiopulmonary resuscitation, pneumonia, unplanned intubation, the need for a mechanical ventilator for ≥48 h after surgery, deep venous thrombosis, pulmonary embolism, stroke, acute kidney injury, progressive renal insufficiency, deep SSI, organ space infection, blood transfusion, wound disruption, and systemic sepsis. Further assessed were the length of stay (LOS) (index surgery) and prolonged hospitalization defined as LOS > 12 days (3rd quartile for LOS of the whole cohort). In addition, unplanned readmission, unplanned reoperation and 30 day mortality were reported.

### 2.5. Statistical Analysis

Descriptive statistics were reported as median (interquartile range IQR) for continuous variables and as frequencies and percentages for categorical variables. The differences between the four groups were compared using the chi-squared test for categorical variables and the Independent Samples Kruskal–Wallis test for continuous variables. Outcomes with an alpha level < 0.1 in the univariable analysis were then included in the multivariable binary logistic regression. The odds ratios (OR) with their corresponding 95% confidence intervals (95% CI) are presented. For all analyses, an alpha level < 0.05 was considered statistically significant and all tests were two-sided.

Statistical analysis was conducted using the Statistical Package for the Social Sciences SPSS Advanced Statistics 25 (IBM Software Group, Inc., Armonk, NY, USA).

## 3. Results

### 3.1. Prevalence of Malnutrition

A total of 25,431 patients were identified. Of those, 6560 (25.8%) patients had severe hypoalbuminemia, 380 (1.5%) patients met ESPEN 2 criteria only, and 671 (2.6%) patients met both criteria (severe hypoalbuminemia and ESPEN 2) (Figure 1).

A total of 10,702 patients (42.1%) had UC, while 14,729 (57.9%) had CD. Severe hypoalbuminemia was more prevalent in CD patients (*p* < 0.001) (Figure 2).

#### Prevalence of Malnutrition According to Surgical Setting

Overall, 1905 patients (7.5%) underwent emergency surgery. Emergency surgeries were more prevalent in patients with UC compared to patients with CD (910 patients (8.5%) vs. 995 patients (6.8%); *p* < 0.0001). Malnutrition was more prevalent in patients who underwent emergency surgery compared to patients who had elective surgery, regardless of the underlying disease and the definition used for malnutrition (Figure 3).

### 3.2. Baseline Characteristics

Baseline and surgical characteristics of the four comparative groups are displayed in Table 1.

### 3.3. Outcomes

Patients who presented with malnutrition according to any of the three definitions had higher rates of overall, minor, major, surgical, and medical complications, longer LOS, higher mortality, and higher rates of readmission and reoperation, as specified in Table 2.

After adjusting for baseline characteristics, the patients who had clinical malnutrition had higher adjusted odds of overall, major, surgical and medical complications compared to the patients without severe malnutrition (reference group). The patients who had severe hypoalbuminemia had higher adjusted odds of overall complications and prolonged LOS compared to patients who had clinical malnutrition alone (ESPEN 2). There was no statistically significant difference in the adjusted odds regarding major, surgical and medical complications, or in the overall mortality between patients who had severe hypoalbuminemia compared to patients who had clinical malnutrition alone (ESPEN 2) (Figure 4).

## 4. Discussion

This large scale study confirms both the high prevalence and predictive value of severe preoperative hypoalbuminemia (<3.1 g/L) as a marker of surgical risk in IBD patients. In contrast, simple clinical assessment of malnutrition based on the ESPEN 2 recommendations, combining BMI and weight loss, considerably underestimates the true prevalence of malnutrition, while being strongly associated with postoperative adverse outcomes.

Malnutrition in IBD patients is common with a prevalence of 16%, according to a recent Spanish multicenter study using the Subjective Global Assessment tool and bioelectrical impedance [11]. In surgical IBD patients facing disease complications leading to both malabsorption and nutrient loss, malnutrition affects up to 50% of patients, according to the Global Leadership Initiative on Malnutrition (GLIM) tool [12]. Hence, its prevalence is highly dependent on the used screening modality and a gold standard method is still lacking [13]. Official ESPEN guidelines suggest both BMI and preoperative weight loss as screening tools for preliminary clinical assessment [9]. However, BMI may be an inaccurate measure of body fat composition since it does not take into account muscle mass and bone density among other factors [14]. Assessing body fat composition in the surgical patient may be of importance given the increasing evidence demonstrating a direct relationship between postoperative hospital stay and surgical outcomes [15,16]. Furthermore, BMI scores may remain within normal ranges despite significant preoperative cachexia, especially in the US population facing an increasing prevalence of obesity [17]. The present study confirms these findings, given the very low prevalence of about 4% when using this composite clinical screening tool, which may lead to the underdetection of patients at risk. The assessment of body composition through anthropometric measures may be more accurate and indispensable in this setting [18,19]. Furthermore, validated official nutritional screening tools such as the Nutritional Risk Score (NRS-2002), the Malnutrition Universal Screening Tool (MUST) or (GLIM) need to be considered, together with dedicated assessment by nutritional specialists [12,20,21].

Hypoalbuminemia has been repeatedly described as a risk factor for postoperative adverse outcomes and, in particular, intraabdominal septic complications in surgical IBD patients [5,22]. Preoperative albumin levels correlate with systemic inflammation and thus, reflect disease severity. Hence, albumin represents a surrogate of disease activity rather than a marker of malnutrition. This is further supported by the recently published position paper of the American Society for Parenteral and Enteral Nutrition (ASPEN) [23]. According to these guidelines, serum albumin must be recognized as an inflammatory marker associated with “nutrition risk” in the context of nutrition assessment, rather than with malnutrition per se. Furthermore, serum albumin does not serve as a valid proxy measure of total body protein or total muscle mass. Serum albumin, as an acute phase reactant in proinflammatory states, decreases as a result of increased vessel permeability, increased clearance, hepatic repriorization, and alterations in liver synthesis [23,24]. Importantly, the present study tried to adjust for unrelated causes of hypoalbuminemia by excluding patients with underlying hepatopathy. The high prevalence of severe hypoalbuminemia (< 3.1 g/L) of over 25% in this cohort of all-comers underlines the fragility of surgical IBD patients and calls for preoperative optimization strategies in this vulnerable patient population, including the correction of anemia, weaning of steroids, the treatment of malnutrition, and intraabdominal infection control among others [25]. The present study further suggests a cumulative deleterious effect of both hypoalbuminemia and clinical malnutrition, which may be even more pronounced in the emergency setting. As a consequence, patients should benefit from a global conditioning concept.

The preconditioning of surgical IBD patients is mandatory to achieve better surgical outcomes [3]. Specific recommendations to face preoperative anemia have been described by the European Crohn’s and Colitis Organization (ECCO) [26]. Medical optimization (i.e., steroid weaning) and the treatment of malnutrition represent further strategies to facilitate and improve surgical management [27,28]. Preoperative nutritional supplementation strategies in patients at risk have been recommended by the ESPEN in dedicated guidelines and need to be tailored to the individual patient and risk profile. As a common denominator, enteral support strategies should be preferred over parenteral nutrition whenever possible [3,13]. The present study emphasizes the importance of these optimization strategies in the light of increased postoperative surgical, medical and infectious complications associated with severe hypoalbuminemia and malnutrition.

This study has limitations associated but not exclusively linked to the ACS-NSQIP, with its unselected 20% sample of the surgical US population. First, the comparative baseline group not fulfilling any of the aforementioned criteria may have met other criteria for malnutrition not accounted for. Second, the chosen definition, ESPEN 2, relies on a previously described definition [9]. However, both the ESPEN and the ASPEN endorse further screening tools (i.e., muscle mass, lean body mass) which were not available in this dataset [13,23]. Third, the large scale of this study impedes in-depth analysis of individual institutional practices regarding nutritional screening and therapy, which has to be considered when interpreting the results. Finally, albumin as an unspecific marker of disease activity should not replace dedicated nutritional screening but may help to assess surgical risk in a busy clinical practice.

## 5. Conclusions

In conclusion, this analysis revealed a low prevalence of malnutrition when defined as a low BMI in conjunction with significant weight. In order to tailor preoperative support strategies and to identify and treat patients at nutrition risk, nutritional screening through validated scores and referral to dedicated specialists to implement preoperative support should be strongly advocated. Albumin as an unspecific marker of disease activity may help to define whether preoperative support strategies are needed.

## Figures and Tables

**Figure 1 nutrients-14-00932-f001:**
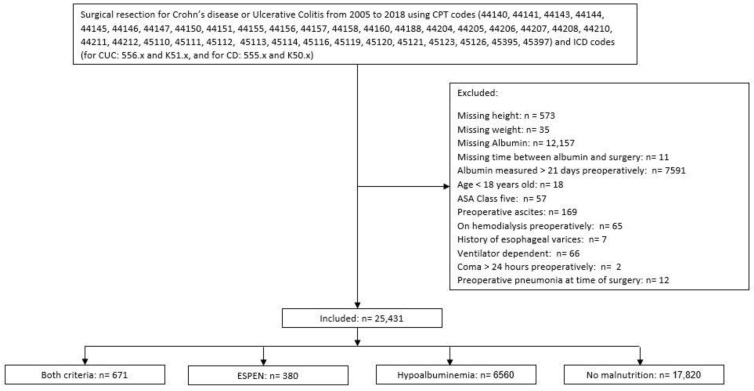
Study Flow Diagram. ASA: American Society of Anesthesiologists, ESPEN 2: criteria according to the European Society for Nutrition and Metabolism [9], hypoalbuminemia: albumin level < 3.1 g/L.

**Figure 2 nutrients-14-00932-f002:**
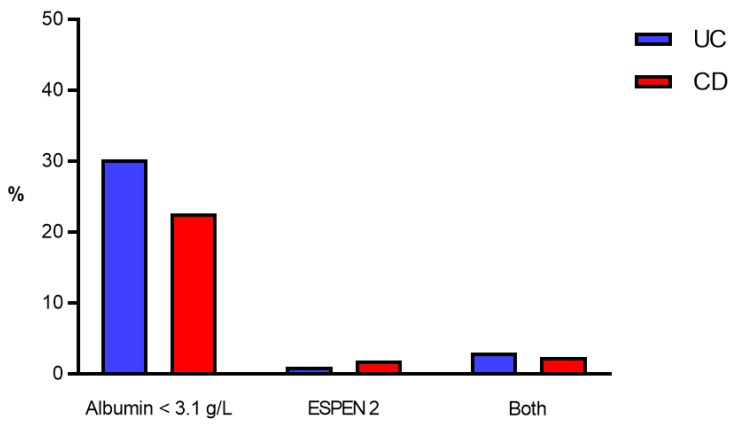
Prevalence of malnutrition criteria. Prevalence of malnutrition criteria (in %) in UC patients (blue bars) and CD patients (red bars). ESPEN 2—clinical criteria according to the European Society for Nutrition and Metabolism [9]. CD—Crohn’s disease, UC—Ulcerative colitis.

**Figure 3 nutrients-14-00932-f003:**
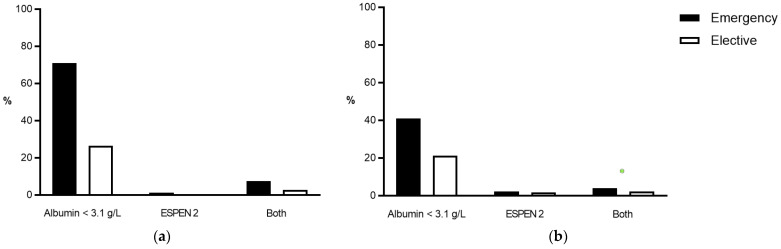
Prevalence of malnutrition criteria according to surgical setting. Comparison of the prevalence of malnutrition (in %) before emergency (black bars) and elective operations (white bars) in (**a**) UC and (**b**) CD patients. ESPEN 2—clinical criteria according to the European Society for Nutrition and Metabolism. CD—Crohn’s disease, UC—Ulcerative colitis.

**Figure 4 nutrients-14-00932-f004:**
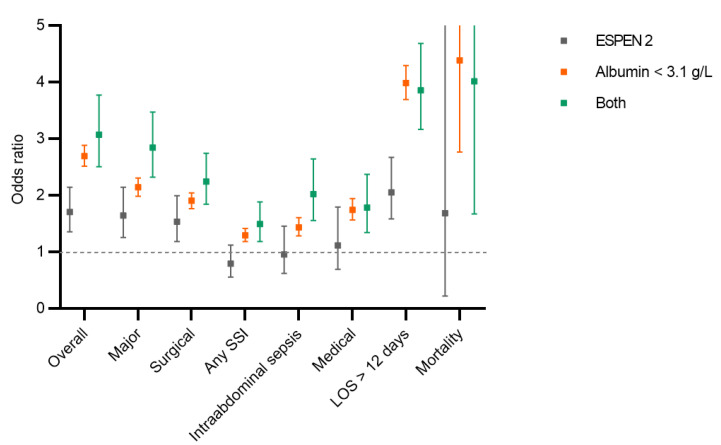
Postoperative complications. Comparison of different postoperative complications, prolonged length of hospital stay and mortality between patients with hypoalbuminemia, ESPEN 2 criteria [9] and both criteria. All complications were adjusted for demographic and surgical risk factors. Displayed are odds ratios (squares) with 95% confidence interval. None of the three criteria met was used as a reference. SSI: surgical site infection, LOS: length of stay.

**Table 1 nutrients-14-00932-t001:** Baseline characteristics.

	Both Criteria *n* = 671	ESPEN 2 *n* = 380	Hypoalbuminemia *n* = 6560	No Malnutrition *n* = 17,820	Total *n* = 25,431	*p* Value
Age ≥ 40 years	286 (42.6%)	113 (29.7%)	3678 (56.1%)	9163 (51.4%)	13,240 (52.1%)	<0.0001
BMI	17.2 (16.2–18.3)	17.6 (16.6–18.4)	23.4 (20.5–27.5)	25.2 (21.9–29.4)	24.4 (21–28.7)	<0.0001
Sex						0.038
Male	376 (56.0%)	203 (53.6%)	3344 (51.0%)	9034 (50.7%)	12,957 (51.0%)	
Race						<0.0001
White	473 (70.5%)	298 (78.4%)	5078 (77.4%)	14,442 (81.0%)	20,291 (79.8%)	
African American	70 (10.4%)	34 (8.9%)	613 (9.3%)	1251 (7.0%)	1968 (7.7%)	
Asian	20 (3.0%)	6 (1.6%)	94 (1.4%)	244 (1.4%)	364 (1.4%)	
Others	4 (0.6%)	4 (1.1%)	18 (0.3%)	71 (0.4%)	97 (0.4%)	
Missing or unknown	104 (15.5%)	38 (10.0%)	757 (11.5%)	1812 (10.2%)	2711 (10.7%)	
ASA class						<0.0001
≥3	432 (64.4%)	144 (38.0%)	3900 (59.5%)	6755 (37.9%)	11,231 (44.2%)	
DM	19 (2.8%)	1 (0.3%)	494 (7.5%)	887 (5.0%)	1401 (5.5%)	<0.0001
Current smoker	128 (19.1%)	71 (18.7%)	1146 (17.5%)	3150 (17.7%)	4495 (17.7%)	0.720
Dyspnea	43 (6.4%)	6 (1.6%)	270 (4.1%)	494 (2.8%)	813 (3.2%)	<0.0001
History of COPD	20 (3.0%)	4 (1.1%)	161 (2.5%)	278 (1.6%)	463 (1.8%)	<0.0001
Functional status						<0.0001
Dependent	33 (4.9%)	4 (1.1%)	310 (4.7%)	140 (0.8%)	487 (1.9%)	
CHF within 30 days of surgery	7 (1.0%)	0	44 (0.7%)	19 (0.1%)	70 (0.3%)	<0.0001
HTN requiring medications	54 (8.0%)	26 (6.8%)	1292 (19.7%)	3137 (17.6%)	4509 (17.7%)	<0.0001
Disseminated cancer	9 (1.3%)	1 (0.3%)	45 (0.7%)	50 (0.3%)	105 (0.4%)	<0.0001
Wound class						<0.0001
Clean or clean contaminated	290 (43.2%)	185 (48.7%)	3229 (49.2%)	12,234 (68.7%)	15,938 (62.7%)	
Contaminated or dirty	381 (56.8%)	195 (51.3%)	3331 (50.8%)	5586 (31.3%)	9493 (37.3%)	
> 10% loss of body weight in last 6 months	671 (100.0%)	380 (100.0%)	1206 (18.4%)	937 (5.3%)	3194 (12.6%)	<0.0001
Chronic steroid use	468 (69.7%)	258 (67.9%)	4125 (62.9%)	8945 (50.2%)	13,796 (54.2%)	<0.0001
Bleeding disorder	49 (7.3%)	9 (2.4%)	449 (6.8%)	407 (2.3%)	914 (3.6%)	<0.0001
pRBC transfusion within 72 h before surgery	92 (13.7%)	3 (0.8%)	628 (9.6%)	129 (0.7%)	852 (3.4%)	<0.0001
Underlying disease						<0.0001
UC	319 (47.5%)	109 (28.7%)	3229 (49.2%)	7045 (39.5%)	10,702 (42.1%)	
CD	352 (52.5%)	271 (71.3%)	3331 (50.8%)	10,775 (60.5%)	14,729 (57.9%)	
MIS	306 (45.6%)	188 (49.5%)	2735 (41.7%)	9145 (51.3%)	12,374 (48.7%)	<0.0001
Extent of resection						<0.0001
Segmental colectomy	292 (43.5%)	232 (61.1%)	2946 (44.9%)	9435 (52.9%)	12,905 (50.7%)	
Total colectomy	278 (41.4%)	90 (23.7%)	2494 (38.0%)	2598 (14.6%)	5460 (21.5%)	
Proctocolectomy	87 (13.0%)	32 (8.4%)	895 (13.6%)	3054 (17.1%)	4068 (16.0%)	
Ostomy/others	1 (0.1%)	7 (1.8%)	66 (1.0%)	266 (1.5%)	340 (1.3%)	
Proctectomy	13 (1.9%)	19 (5.0%)	159 (2.4%)	2467 (13.8%)	2658 (10.5%)	
Diversion at time of index surgery	445 (66.3%)	164 (43.2%)	4268 (65.1%)	8666 (48.6%)	13,543 (53.3%)	<0.0001
Urgency of surgery						<0.0001
Emergent	107 (15.9%)	25 (6.6%)	1053 (16.1%)	720 (4.0%)	1905 (7.5%)	
Sepsis or septic shock at time of surgery						<0.0001
Yes	46 (6.9%)	11 (2.9%)	382 (5.8%)	264 (1.5%)	703 (2.8%)	
Missing	143 (21.3%)	83 (21.8%)	1477 (22.5%)	3870 (21.7%)	5573 (21.9%)	
Organ space infection at time of surgery						<0.0001
Yes	42 (6.3%)	8 (2.1%)	267 (4.1%)	216 (1.2%)	533 (2.1%)	
Missing	136 (20.3%)	79 (20.8%)	1439 (21.9%)	3810 (21.4%)	5464 (21.5%)	
Operation time	155 (117–292)	150 (108–208)	177 (129–241)	181 (129–252)	179 (128–248)	<0.0001
Days between albumin measurement and operation	2 (0–5)	4 (1–8)	2 (0–5)	6 (2–11)	4 (1–9)	<0.0001
Preoperative hematocrit	30 (27–33)	35 (31–39)	31 (28–35)	39 (35–42)	37 (32–41)	<0.0001
Preoperative platelets	363 (278–468)	343 (273–456)	341 (255–441)	299 (239–374)	310 (243–394)	<0.0001
Preoperative SGOT	14 (11–22)	15 (12–22)	15 (11–22)	19 (14–25)	18 (13–24)	<0.0001
Preoperative INR	1.1 (1.01–1.23)	1.1 (1–1.2)	1.1 (1.02–1.22)	1.01 (1–1.1)	1.1 (1–1.2)	<0.0001

ESPEN 2: criteria according to the European Society for Nutrition and Metabolism, BMI: body mass index, ASA: American Society of Anesthesiologists, DM: diabetes mellitus, COPD: chronic obstructive pulmonary disease, CHF: congestive heart failure, HTN: hypertension, pRBC: packed red blood cells, UC: ulcerative colitis, CD: Crohn’s disease, MIS: Minimally invasive surgery, Hct: hematocrit, SGOT: glutamic oxaloacetic transaminase, INR: international normalized ratio.

**Table 2 nutrients-14-00932-t002:** Unadjusted complications rates.

	Both Criteria *n* = 671	ESPEN 2 *n* = 380	Hypoalbuminemia *n* = 6560	No Malnutrition *n* = 17,820	Total *n* = 25,431	*p* Value
Overall complications	484 (72.1%)	188 (49.5%)	4308 (65.7%)	5996 (33.6%)	10,976 (43.2%)	<0.0001
Minor complications	46 (6.9%)	25 (6.6%)	623 (9.5%)	1362 (7.6%)	2056 (8.1%)	<0.0001
Major complications	319 (47.5%)	100 (26.3%)	2668 (40.7%)	3260 (18.3%)	6347 (25.0%)	<0.0001
Surgical complications	296 (44.1%)	105 (27.6%)	2568 (39.1%)	3528 (19.8%)	6497 (25.5%)	<0.0001
Any SSI	139 (20.7%)	40 (10.5%)	1213 (18.5%)	2322 (13.0%)	3714 (14.6%)	<0.0001
Superficial	27 (4.0%)	15 (3.9%)	413 (6.3%)	998 (5.6%)	1453 (5.7%)	0.013
Deep	9 (1.3%)	3 (0.8%)	116 (1.8%)	238 (1.3%)	366 (1.4%)	0.059
Organ/Space	109 (16.2%)	27 (7.1%)	762 (11.6%)	1210 (6.8%)	2108 (8.3%)	<0.0001
Wound disruption	9 (1.3%)	4 (1.1%)	129 (2.0%)	140 (0.8%)	282 (1.1%)	<0.0001
Systemic sepsis	89 (13.3%)	31 (8.2%)	731 (11.1%)	980 (5.5%)	1831 (7.2%)	<0.0001
Sepsis	89 (13.3%)	31 (8.2%)	731 (11.1%)	980 (5.5%)	1831 (7.2%)	<0.0001
Septic shock	23 (3.4%)	4 (1.1%)	280 (4.3%)	161 (0.9%)	468 (1.8%)	<0.0001
Need for blood transfusion	187 (27.9%)	52 (13.7%)	1506 (23.0%)	1105 (6.2%)	2850 (11.2%)	<0.0001
Medical complications	89 (13.3%)	21 (5.5%)	1007 (15.4%)	1178 (6.6%)	2295 (9.0%)	<0.0001
UTI	19 (2.8%)	10 (2.6%)	243 (3.7%)	423 (2.4%)	695 (2.7%)	<0.0001
Respiratory complications	42 (6.3%)	8 (2.1%)	464 (7.1%)	324 (1.8%)	838 (3.3%)	<0.0001
Renal complications	4 (0.6%)	2 (0.5%)	114 (1.7%)	159 (0.9%)	279 (1.1%)	<0.0001
MACE	9 (1.3%)	2 (0.5%)	87 (1.3%)	54 (0.3%)	152 (0.6%)	<0.0001
VTE	32 (4.8%)	2 (0.5%)	356 (5.4%)	376 (2.1%)	766 (3.0%)	<0.0001
LOS, Median (IQR), days	13 (8–20)	8 (5–14)	12 (7–19)	6 (4–9)	7 (4–12)	<0.0001
LOS > 12 day (Q3)	330 (49.7%)	98 (25.8%)	2943 (45.2%)	2078 (11.7%)	5449 (21.5%)	<0.0001
Unplanned readmission related to the principle procedure *	88 (16.7%)	50 (16.8%)	769 (15.1%)	1875 (13.4%)	2782 (14.0%)	0.003
Unplanned reoperation related to the principle procedure *	58 (11.0%)	17 (5.7%)	379 (7.5%)	647 (4.6%)	1101 (5.5%)	<0.0001
Thirty day mortality	11 (1.6%)	1 (0.3%)	125 (1.9%)	29 (0.2%)	166 (0.7%)	<0.0001

ESPEN 2: criteria according to the European Society for Nutrition and Metabolism [9], SSI: surgical site infection, UTI: urinary tract infection, MACE: major adverse cardiovascular events, VTE: vascular thromboembolism, LOS: length of stay, IQR: interquartile range, Q3: third quartile. * Data only available for the years after 2012.

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
