# Peer review of "Simple Clinical Screening Underestimates Malnutrition in Surgical Patients with Inflammatory Bowel Disease—An ACS NSQIP Analysis"

_nutrients, 2022, doi:10.3390/nu14050932_

Round 1
Reviewer 1 Report
The major and fundamental flaw in this study is visceral proteins (such as albumin, prealbumin, and similar proteins) are just indicative of the severity of the disease and inflammation and do NOT indicate nutritional status.
I encourage authors to consider reading the recent ASPEN position paper on this topic (Evans DC, Corkins MR, Malone A, Miller S, Mogensen KM, Guenter P, Jensen GL; ASPEN Malnutrition Committee. The Use of Visceral Proteins as Nutrition Markers: An ASPEN Position Paper. Nutr Clin Pract. 2021 Feb;36(1):22-28. doi: 10.1002/ncp.10588. Epub 2020 Oct 30. Erratum in: Nutr Clin Pract. 2021 Aug;36(4):909. PMID: 33125793) which nicely discusses this limitation.
Given the challenges in malnutrition assessment, unfortunately, clinicians still consider albumin (which is readily available) as a surrogate for nutritional assessment. Specifically, in IBD - hypoalbuminemia is very likely due to protein loss from enteropathy/increased capillary permeability, and hepatic reprioritization in the production of acute-phase reactants than due to malnutrition.
The low prevalence of malnutrition based on ESPEN criteria is just reflective of the limitation of this retrospective study design and underrecognition of malnutrition among clinicians which is well documented in various disease populations.
Author Response
Reviewer #1: The major and fundamental flaw in this study is visceral proteins (such as albumin, prealbumin, and similar proteins) are just indicative of the severity of the disease and inflammation and do NOT indicate nutritional status.
I encourage authors to consider reading the recent ASPEN position paper on this topic (Evans DC, Corkins MR, Malone A, Miller S, Mogensen KM, Guenter P, Jensen GL; ASPEN Malnutrition Committee. The Use of Visceral Proteins as Nutrition Markers: An ASPEN Position Paper. Nutr Clin Pract. 2021 Feb;36(1):22-28. doi: 10.1002/ncp.10588. Epub 2020 Oct 30. Erratum in: Nutr Clin Pract. 2021 Aug;36(4):909. PMID: 33125793) which nicely discusses this limitation.
Given the challenges in malnutrition assessment, unfortunately, clinicians still consider albumin (which is readily available) as a surrogate for nutritional assessment. Specifically, in IBD - hypoalbuminemia is very likely due to protein loss from enteropathy/increased capillary permeability, and hepatic reprioritization in the production of acute-phase reactants than due to malnutrition.
The low prevalence of malnutrition based on ESPEN criteria is just reflective of the limitation of this retrospective study design and underrecognition of malnutrition among clinicians which is well documented in various disease populations.
Authors' response: We agree with the reviewer and do not claim hypoalbuminemia being a marker of malnutrition but rather a surrogate of disease severity, especially in the CD population. We tried to be more specific about this point and amended the manuscript throughout. We kindly refer to the following paragraph, which we adapted:
Hypoalbuminemia has been repeatedly described as a risk factor for postoperative adverse outcomes and in particular intraabdominal septic complications in surgical IBD patients. Preoperative albumin levels correlate with systemic inflammation and thus reflect disease severity. Hence, albumin represents a surrogate of disease activity rather than a true marker of malnutrition. Serum albumin as an acute phase reactant in proinflammatory states decreases as a result of increased vessel permeability, increased clearance hepatic repriorization and alterations in liver synthesis.
We amended the paragraph further by adding the following referenced statement:
This is further supported by the recently published position paper of the American Society for Parenteral and Enteral Nutrition (ASPEN). According to these guidelines, serum albumin must be recognized as inflammatory marker associated with “nutrition risk” in the context of nutrition assessment rather than with malnutrition per se. Furthermore, serum albumin does not serve as valid proxy measure of total body protein or total muscle mass.
As pointed out by the reviewer, underrecognition of malnutrition among clinicians (and in particular surgeons) represents a major issue. We believe that, despite the limitations related to the large dataset and the retrospective character, the main message is in line with the reviewers’ point of view. Underrecognition of malnutrition by clinicians/surgeons, albumin as a marker of disease activity/severity and thus surgical risk, importance of proper nutiritonal assessment beyond clinical assessment.
We amended several statements throughout the manuscript including the conclusion in line with the reviewers’ suggestions, which helped us to improve our manuscript (highlighted in yellow).
Reviewer 2 Report
Paper "Simple clinical screening underestimates malnutrition in surgical patients with inflammatory bowel disease – an ACS NSQIP 3 analysis" demonstrated in the large cohort of inflammatory bowel disease patients the limitations of currently applied tools to identify individuals with malnutrition. Additionally, the authors performed the study on the patients undergoing surgery, which constitute a group where the proper assessment of the nutritional status is significant and still underestimated. Therefore, the study may have important clinical implications. The manuscript not only shed new light on the utility of the recommended tests and serum albumin as a biomarker but also confirmed the correlation between the nutritional status and the surgical complications.
Author Response
We would like to thank the reviewer for the encouraging comments.
Reviewer 3 Report
This paper presents data from a large database of US patients with IBD who had surgery. The paper uses this data to retrospectively assess risk of surgical complications with nutrition status using part of the ESPEN malnutrition criteria and/or hypoalbuminemia. Data on inflammatory burden and the correlation with albumin is not clearly articulated or presented. This needs to be addressed as it is well documented that albumin is a non-specific marker of disease activity and not a marker of nutritional status during active IBD.
Specific comments:
Line 34 The first paragraph of the introduction could be improved. The first few sentences are contradictory, for example it says that elective surgery is frequently not an option but then the following sentence suggests that preoperative nutrition support is needed. If elective is not that common how can there be adequate time to “launch preoperative support”? Suggest revising the word “impeding”. The next sentence introduces the idea of hypoalbuminemia being a marker of malnutrition but it is well documented that hypoalbuminemia is driven by inflammation.
Line 39 - How applicable is the ESPEN criteria for malnutrition to the US population? What proportion of the IBD population in the US have a BMI less 22 kg/m2 (10% weight loss would then result in a BMI less than 20 kg/m2). The focuses on discrediting the ESPEN criteria but does not introduce why these criteria may not be an appropriate screening tool in the US.
Section 2.3 covariates and outcomes. Inflammatory burden (other than platelets) should be considered as a covariate. In the inflammatory setting albumin is not a marker of nutritional status but a marker of inflammation https://aspenjournals.onlinelibrary.wiley.com/doi/10.1002/jpen.1451 . It is well documented that preoperative inflammatory burden is an independent risk factor for surgical complications in IBD.
Line 163 – completely agree with this conclusion.
Line 166 – this study did not assess BMI or weight loss only, it assessed low BMI AND weight loss. The study did not assess whether weight loss only (regardless of BMI) is associated with adverse postoperative outcomes. Please revise.
Line 168 – how did the Spanish study (reference 10) and reference 11 define malnutrition? How was it different from the current study that had a much lower prevalence of malnutrition?
Line 174 – why does body fat composition matter in the surgical patient? Please add a sentence to explain how this relevant.
Line 185 onwards – the first half of this paragraph is great.
Line 194 – 195 – make it clear what type of preoperatively strategies are being referred to here. Nutrition strategies that can reduce inflammatory burden? Based on Figure 4 it appears that there is a much higher odds of surgical complications in patients with high inflammatory burden coupled with low BMI and significant weight loss but just low BMI without high inflammatory burden (hypoalbuminemia) not as much.
Lines 200 – 203 – suggest removing these sentences as micronutrient status is not reported in the results.
Line 221 – 223 – albumin is not a marker of nutritional status in patients with active IBD and should not be used as such in a busy clinical practice. https://aspenjournals.onlinelibrary.wiley.com/doi/10.1002/jpen.1451
Line 226 – please revise to reflect the criteria applied in the paper. E.g. “revealed a low prevalence of malnutrition when defined as low BMI in conjunction with significant weight loss.”.
Line 227 – suggest adding “nutrition” before the word risk.
Line 228 – the point of a screening tool is that it can be completed by range of healthcare providers. Suggest revising the last part of the sentence to say e.g. and referral to dedicated specialists to implement preoperative support should strongly be advocated.
Line 228 – suggest making the final sentence more specific because at the moment it is open to wide interpretation. For example, “Albumin, as an unspecific marker of disease activity, may help to refine whether inflammation targeted or nutrition supportive preoperative support strategies are needed.
Author Response
This paper presents data from a large database of US patients with IBD who had surgery. The paper uses this data to retrospectively assess risk of surgical complications with nutrition status using part of the ESPEN malnutrition criteria and/or hypoalbuminemia. Data on inflammatory burden and the correlation with albumin is not clearly articulated or presented. This needs to be addressed as it is well documented that albumin is a non-specific marker of disease activity and not a marker of nutritional status during active IBD.
Thank you for this comment, in line with the suggestion of reviewer 1. Please find detailed explanations and changes in the manuscript to clarify further this point.
Specific comments:
Line 34 The first paragraph of the introduction could be improved. The first few sentences are contradictory, for example it says that elective surgery is frequently not an option but then the following sentence suggests that preoperative nutrition support is needed. If elective is not that common how can there be adequate time to “launch preoperative support”? Suggest revising the word “impeding”. The next sentence introduces the idea of hypoalbuminemia being a marker of malnutrition but it is well documented that hypoalbuminemia is driven by inflammation.
Both statements were adapted in line with the reviewers’ recommendations.
Line 39 - How applicable is the ESPEN criteria for malnutrition to the US population? What proportion of the IBD population in the US have a BMI less 22 kg/m2 (10% weight loss would then result in a BMI less than 20 kg/m2). The focuses on discrediting the ESPEN criteria but does not introduce why these criteria may not be an appropriate screening tool in the US.
We added a statement to highlight the risks of clinical assessment alone of malnutrition in the US population.
Section 2.3 covariates and outcomes. Inflammatory burden (other than platelets) should be considered as a covariate. In the inflammatory setting albumin is not a marker of nutritional status but a marker of inflammation https://aspenjournals.onlinelibrary.wiley.com/doi/10.1002/jpen.1451 . It is well documented that preoperative inflammatory burden is an independent risk factor for surgical complications in IBD.
This is an important comment. Hypoalbuminemia was clearly associated with increased complications of any kind – even more when paired with clinical malnutrition (ESPEN 2). This reflects the increased surgical risk due to systemic inflammation. Further inflammatory markers such as C-reactive protein were unfortunately not available.
In order to compare the groups (Fig. 4), we chose to create mutually exclusive groups, which show the increasing risks depending on the group.
Line 163 – completely agree with this conclusion.
Line 166 – this study did not assess BMI or weight loss only, it assessed low BMI AND weight loss. The study did not assess whether weight loss only (regardless of BMI) is associated with adverse postoperative outcomes. Please revise.
We revised accordingly in line 166 and the conclusion.
Line 168 – how did the Spanish study (reference 10) and reference 11 define malnutrition? How was it different from the current study that had a much lower prevalence of malnutrition?
We added this pertinent information to both statements (SGA and GLIM tools, bioelectrical impedance).
Line 174 – why does body fat composition matter in the surgical patient? Please add a sentence to explain how this relevant.
We gladly added a referenced explanatory statement.
Line 185 onwards – the first half of this paragraph is great.
Line 194 – 195 – make it clear what type of preoperatively strategies are being referred to here. Nutrition strategies that can reduce inflammatory burden? Based on Figure 4 it appears that there is a much higher odds of surgical complications in patients with high inflammatory burden coupled with low BMI and significant weight loss but just low BMI without high inflammatory burden (hypoalbuminemia) not as much.
This is true but also related to the aforementioned underestimation of nutrition risk, which certainly represents an important area of preoperative optimization. We specified the concept of preoperative optimization as suggested. Regarding the next comment, we removed the sentence related to micronutrients as suggested.
Lines 200 – 203 – suggest removing these sentences as micronutrient status is not reported in the results.
Line 221 – 223 – albumin is not a marker of nutritional status in patients with active IBD and should not be used as such in a busy clinical practice. https://aspenjournals.onlinelibrary.wiley.com/doi/10.1002/jpen.1451
We rephrased according to this suggestion and the suggestions of reviewer 1.
Line 226 – please revise to reflect the criteria applied in the paper. E.g. “revealed a low prevalence of malnutrition when defined as low BMI in conjunction with significant weight loss.”
We rephrased according to this precise statement.
Line 227 – suggest adding “nutrition” before the word risk.
The word was added as suggested.
Line 228 – the point of a screening tool is that it can be completed by range of healthcare providers. Suggest revising the last part of the sentence to say e.g. and referral to dedicated specialists to implement preoperative support should strongly be advocated.
Line 228 – suggest making the final sentence more specific because at the moment it is open to wide interpretation. For example, “Albumin, as an unspecific marker of disease activity, may help to refine whether inflammation targeted or nutrition supportive preoperative support strategies are needed.
Thank you for helping us to refine our conclusion and for the valuable input throughout.
Round 2
Reviewer 1 Report
The revised manuscript is better.